# Characterizing clinical findings of Sjögren's Disease patients in community practices using matched electronic dental-health record data

**Grace Gomez Felix Gomez**[1,2], **Steven T. Hugenberg**[3,4], **Susan Zunt**[5], **Jay S. Patel**[1,6], **Mei Wang**[1], **Anushri Singh Rajapuri**[1,2], **Lauren R. Lembcke**[7], **Divya Rajendran**[1,8], **Jonas C. Smith**[7,9], **Biju Cheriyan**[1], **LaKeisha J. Boyd**[10], **George J. Eckert**[10], **Shaun J. Grannis**[2,3], **Mythily Srinivasan**[5], **Domenick T. Zero**[1], **Thankam P. Thyvalikakath**[1,2]*

1 Dental Informatics, Department of Cariology, Operative Dentistry & Dental Public Health, Indiana University School of Dentistry, Indianapolis, Indiana, United States of America, 2 Center for Biomedical Informatics, Regenstrief Institute Inc., Indianapolis, Indiana, United States of America, 3 Indiana University School of Medicine, Indianapolis, Indiana, United States of America, 4 Indiana University Health, Indianapolis, Indiana, United States of America, 5 Department of Oral Pathology, Medicine and Radiology, Indiana University School of Dentistry, Indianapolis, Indiana, United States of America, 6 Department of Health Services Administration and Policy, College of Public Health, Temple University, Philadelphia, Pennsylvania, United States of America, 7 Regenstrief Data Services, Regenstrief Institute Inc, Indianapolis, Indiana, United States of America, 8 Innovation Associates iA, Indianapolis, Indiana, United States of America, 9 Med Shield, Inc., Indianapolis, Indiana, United States of America, 10 Department of Biostatistics and Health Data Sciences, Indiana University School of Medicine, Indianapolis, Indiana, United States of America

* tpt@iu.edu

**Data Availability Statement:** Data cannot be shared publicly because data contain confidential patient information from medical records and are

## Abstract

Established classifications exist to confirm Sjögren's Disease (SD) (previously referred as Sjögren's Syndrome) and recruit patients for research. However, no established classification exists for diagnosis in clinical settings causing delayed diagnosis. SD patients experience a huge dental disease burden impairing their quality of life. This study established criteria to characterize Indiana University School of Dentistry (IUSD) patients' SD based on symptoms and signs in the electronic health record (EHR) data available through the state-wide Indiana health information exchange (IHIE). Association between SD diagnosis, and comorbidities including other autoimmune conditions, and documentation of SD diagnosis in electronic dental record (EDR) were also determined. The IUSD patients' EDR were linked with their EHR data in the IHIE and queried for SD diagnostic ICD9/10 codes. The resulting cohorts' EHR clinical findings were characterized and classified using diagnostic criteria based on clinical experts' recommendations. Descriptive statistics were performed, and Chi-square tests determined the association between the different SD presentations and comorbidities including other autoimmune conditions. Eighty-three percent of IUSD patients had an EHR of which 377 patients had a SD diagnosis. They were characterized as positive (24%), uncertain (20%) and negative (56%) based on EHR clinical findings. Dry eyes and mouth were reported for 51% and positive Anti-Ro/SSA antibodies and anti-nuclear antibody (ANA) for 17% of this study cohort. One comorbidity was present in 98% and other autoimmune condition/s were present in 53% respectively. Significant differences were observed between the three SD clinical characteristics/classifications and certain medical and autoimmune conditions (p<0.05). Sixty-nine percent of patients' EDR did not

managed by the Regenstrief Institute. Data are available from the Indiana University Institutional Data Access / Ethics Committee (contact Regenstrief Data Services via https://www.regenstrief.org/data-request/) for researchers who meet the criteria for access to confidential data.

**Funding:** The corresponding and senior author (TPT) received federal support for this research through the National Institutes of Health/National Institute of Dental and Craniofacial Research (NIH/NIDCR) grants R21 DE027786-02 and 1R56DE029195-01. The website link for the funding agency is https://www.nidcr.nih.gov/. The funders had no role in the study design, data collection, and analysis, decision to publish, or preparation of the manuscript.

**Competing interests:** The authors have declared that no competing interests exist.

mention SD, highlighting the huge gap in reporting SD during dental care. This study of SD patients diagnosed in community practices characterized three different SD clinical presentations, which can be used to generate SD study cohorts for longitudinal studies using EHR data. The results emphasize the heterogenous SD clinical presentations and the need for further research to diagnose SD early in community practice settings where most people seek care.

## Introduction

Sjögren's Disease (SD) (previously referred as Sjögren's Syndrome (SS)) is a chronic autoimmune connective tissue disorder affecting over 4 million Americans, and an estimated 2.5 million remaining undiagnosed [1–4]. Late awareness or delayed diagnosis occur due to the presentation of nonspecific symptoms such as fatigue, muscle pain, slow progression of the disease and lack of consensus in the diagnostic criteria to be applied in clinical settings [5–11]. As a result, progression of the disease significantly impairs the affected persons' quality of life due to tooth loss, corneal scarring, tiredness, pain, and depression [12–14]. Although oral dryness and salivary gland hypofunction are predisposing symptoms that indicate SD and result in huge dental disease burden, it is uncommonly diagnosed in a dental clinic setting [15–20]. In addition, dentists rely on patients' reporting their medical conditions to recognize patients with SD. Patients' delay in recognizing the symptoms of oral dryness and then reporting them during dental care also increases oral health problems.

During the last four decades, researchers and clinicians have jointly developed criteria to establish SD diagnosis and define a homogenous population of patients eligible for clinical trials [21–25]. These criteria include positive serological tests for autoantibodies, abnormal scores for minor salivary gland biopsy, Schirmer's test, ocular surface staining, and salivary flow rate assessment tests for observed or reported symptoms of dry mouth, dry eye, or extra glandular manifestations [21, 22, 24, 25]. However, applying these classification criteria to patients presenting with nonspecific symptoms could be time-consuming and expensive for patients and providers. These tests also require referral to a rheumatologist, ophthalmologist, oral pathologist, or a dentist. In addition, the heterogenous nature of SD makes it difficult and controversial for the classification criteria to be applied in clinical settings imposing a significant barrier to conduct longitudinal studies regarding oral and overall health and treatment outcomes. As a result, physicians may diagnose SD based on subjective symptoms and positive autoantibody tests [26]; yet they may not refer patients to dental providers to confirm the diagnosis through additional objective tests required by the classification criteria such as salivary flow rate, minor salivary gland biopsy. Due to the varying clinical presentations, physicians and dentists need evidence-based studies in diagnosing patients at risk for SD.

Studies during the past decade have utilized International Classification of Disease (ICD) codes for SD from the electronic health record (EHR) and administrative claims data to determine prevalence in the US (United States), Canada and Taiwan [26–31]. ICD-9 and ICD-10 codes for SD are reported to have higher specificity and sensitivity when compared to the final diagnosis recorded in the EHR [27, 28]. In the first US population-based study using EHR data, the authors assessed the prevalence of physician-diagnosed primary SS in Olmsted County, Minnesota by using the diagnostic codes for SS (referred as Sjogren's disease (SD) in this paper), sicca syndrome, and keratoconjunctivitis sicca (KCS) and then ascertaining the presence of symptoms and diagnostic test results (as described in the 2002 American European

Consensus Group (AECG) criteria) in the EHR [26, 30]. They found that physicians used serological tests, such as anti-SSA (Ro)/anti-SSB (La) antibodies, rheumatoid (RF) factor and antinuclear antibody (ANA), frequently to diagnose primary SS. Moreover, objective exploration of sicca symptoms was rarely performed [26]. To the best of our knowledge, no other study has characterized and classified the clinical findings and tests that US physicians, including rheumatologists, apply to diagnose SD in clinical practices. Also, retrospective analysis of EHR data can investigate the longitudinal outcomes of SD patients. Therefore, more studies characterizing physicians' diagnosis of SD in community settings are crucial to generate such retrospective study cohorts. Moreover, no studies have investigated SD by linking electronic dental record (EDR) with EHR data despite the huge dental disease burden that occur due to SD.

In routine dental practice, the provider relies mostly on patient-reported SD status unless they observe clinically evident dry mouth or salivary gland enlargement, which may occur late in the disease process [32–35]. As a result, limited longitudinal studies exist that characterize this population's oral disease changes, treatment outcomes and the factors influencing these outcomes. As the existing AECG criteria [21, 22] incorporate only the complete manifestation of the disease, and since they are not particularly relevant to detect SD in clinical practice, it is necessary to characterize these patients' clinical characteristics using EHR data for research studies. The digitization of dental care data and the establishment of regional and state-wide health information exchanges (HIE), with increased interoperability between EHRs, offer the opportunity to link EDR and EHR data to identify a larger cohort of SD patients and study their oral health [36, 37]. Therefore, in this study, we matched and linked the Indiana University School of Dentistry's (IUSD) patients' EDR data with their EHR data available through the state-wide Indiana Health Information Exchange (IHIE).

This study aimed to 1) establish a set of criteria that can be operationalized to characterize and classify clinical presentations of patients diagnosed with SD in clinical practice for research purposes; 2) characterize the symptoms and objective findings recorded in the EHR for patients with a SD diagnostic code; 3) determine the association between SD diagnosis and comorbidities experienced by patients; and 4) determine the difference in the documentation of patient-reported SD diagnosis in the EDR versus SD diagnosis recorded in their EHR. The outcomes of this study will establish a set of criteria that can be utilized to generate a cohort of SD diagnosed patients using EHR data, and different clinical presentations facilitating longitudinal studies, determine combination of symptoms and test results used to diagnose SD and the extent to which patients report their SD status while seeking dental care.

## Methods

The study protocol received exempt approval (#1908582138) from the Indiana University (IU) Institutional Review Board (IRB) with the waiver of informed consent from patients. This study utilized a limited data set (containing date of birth and date of treatments) consisting of EDR data of patients who received dental care at IUSD and their EHR data available through the Indiana Network for Patient Care-Research (INPC-R) database maintained by Regenstrief Institute, Inc., and IHIE, a mature state-wide community HIE [38]. The IHIE connects 123 hospitals representing 38 hospital systems, over 19,157 practices, and over 54,505 providers containing approximately 20+ million patients with a total of more than 16 billion data elements [39, 40]. We have successfully matched 83% of IUSD patients' EDR data with their EHR data through INPC-R using the global matching algorithm [41–43]. From the matched patient records that spanned 15 years, selected EDR and EHR data containing study variables were retrieved from records that had a final diagnostic code for SD in their EHR.

## Study population, variables, and covariates

Patients aged 18 years and older with a record of at least one completed dental treatment between January 1, 2005, and December 31, 2020, were included. These patients' EHR were retrieved from the INPC-R database and queried for the presence of International Classification of Diseases, 9th and 10th revision, Clinical Modification (ICD9/10, CMICD9/10, CM) diagnostic codes for SD (710.2; M35.0 to M35.04, M35.09) as well as internal/local concept codes for SD (8232) used by the Regenstrief Data Services (RDS). The resulting cohort included linked EDR-EHR data with at least one occurrence of any of the ICD9/10 or internal codes for SD in the patient's records. Patient charts with the following conditions were excluded: history of head and neck radiation treatment (includes thyroid cancer patients who underwent radiation/ablation with radioactive iodine (mCi), hepatitis C, human immunodeficiency virus/acquired immunodeficiency syndrome (HIV/AIDS), sarcoidosis, pre-existing lymphoma, amyloidosis, graft versus host disease, primary biliary cirrhosis, and Immunoglobulin G4 (IgG4)-related disease. The above mentioned conditions were excluded because they mimic clinical symptoms and other characteristic features of SD. We retrieved demographic and socioeconomic information (date of birth, treatment date, sex, race, ethnicity, and insurance), selected laboratory records, clinical notes, and diagnostic codes (ICD9/10) from the EHR and demographics, sociodemographic information, completed dental treatments, medical history, and clinical notes from the EDR. Fig 1 illustrates an overview of the research study design and procedures.

## Criteria and classification characterizing clinical presentations of patients with SD ICD9/10 codes in the EHR

We developed guidelines based on literature review, rheumatology and oral medicine/oral pathology experts' feedback to classify the patients' SD status as positive, uncertain or negative (see Table 1). A set of criteria was put forth combining the characteristic clinical findings as diagnosed by physicians and features of SD patients recorded in their EHR. The study cohort had a diagnosis of SD in their EHR. Classification was done based on the available symptoms

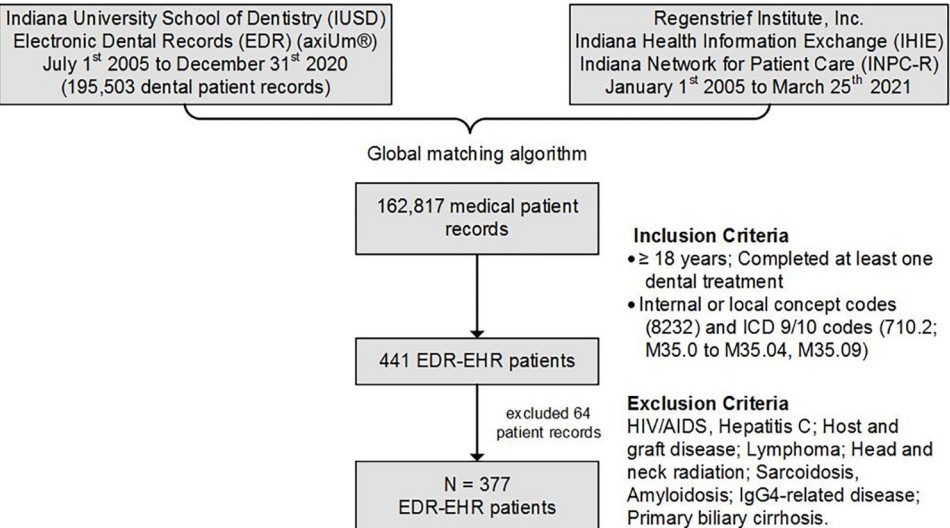

**Fig 1. Study design to match EDR and EHR data and generate SD study cohort.** EDR: electronic dental record; EHR: electronic health record; SD: Sjögren's Disease; ICD 9/10-codes: International Classification of Diseases Codes.

**Table 1. Manual review guidelines to classify patients Sjögren's Disease clinical characteristics using EHR data.**

| Criteria classifying Sjögren's disease patients' clinical characteristics | | | |
|---|---|---|---|
| **Diagnostic tests** | **Positive** | **Uncertain** | **Negative** |
| **Objective findings** | | | |
| **Histopathology** <br> *minor/labial salivary gland biopsy ≥1 foci/4mm² <br> **Serological tests** <br> *autoantibodies–Anti-Ro/SSA) ≥1 IU/ml or/and <br> autoantibodies–Anti-La/SSB ≥1 or <1 IU/ml anti-nuclear antibody (ANA)≥1:160 rheumatoid factor (RF) ≥15 <br> **Ocular sign tests** <br> *ocular stain (OST) test≥3 score <br> **Oral sign tests** <br> *unstimulated salivary flow rate (SFT) test <0.1ml/minute | at least *one positive objective finding of: labial salivary gland biopsy ≥1 focus/4mm² <br> • OR- Anti-Ro/SSA 1 IU/ml <br> • OR- OST test≥3 score <br> • OR- SFT test <0.1 ml/minute | at least #one abnormal serum autoantibody except for positive Anti-Ro/SSA: <br> • Anti-La/SSB ≥1IU/ml (or/and) <br> • (ANA)≥1:160(or/and) <br> • (RF) ≥15; mention of positive RF | Normal objective findings or no available information on diagnostic tests in patient records or presence of positive objective findings with absence of symptoms |
| **(and) Subjective symptoms** | | | |
| dryness of mouth (xerostomia) or documentation of medication/s to manage dry mouth <br> dryness of eyes (keratoconjunctivitis sicca) or documentation of medication/s to manage dry eyes <br> parotid gland enlargement (PGE) | Presence of dry mouth or dry eye or PGE | Presence of dry mouth or dry eye or PGE | Presence/absence of dry mouth or dry eye or PGE |

*One positive objective finding required to be present to confirm Sjögren's Disease diagnosis. #Patient records with positive serological values except for an abnormal Anti-SSA (Ro) are classified uncertain. EHR: electronic health record; Anti-SSA (Ro)–antibody to SSA (Ro) antigen; Anti-SSB (La)—antibody to SSB (La) antigen; RF–Rheumatoid factor; ANA–Anti-nuclear antibody; PGE–Parotid gland enlargement; OST—Ocular staining test; SGB–Salivary gland biopsy; SFT–salivary flow test.

and objective tests present in the EHR data. Clinical notes, selected laboratory results and diagnostic codes present in the EHR data through INPC-R for the study cohort were reviewed manually. The clinical notes resided in nDepth™, and structured data such as laboratory results and diagnostic codes in the IU supported Research Electronic Data Capture (REDCap IU) platform. nDepth™ is a tool that incorporates text mining and natural language processing (NLP) capabilities and assists with exploring and retrieving specific text information through clinical notes. In this study, the terms Sjogren's, dry eyes, dry mouth, SD-specific objective test names, medication names for dry mouth, dry eye, medical condition names belonging to exclusion criteria and their spelling and terminology variations were retrieved by the nDepth tool, through its text mining capabilities. In addition, the nDepth tool highlighted these terms that made the review of clinical notes for relevant information easier for the reviewers.

Finally, thirteen questions (S1 Table; provided in the S1 File) were uploaded into the nDepth's validation module for the reviewers to enter their findings from each patient case. The questions included validating: 1) specific exclusion conditions, 2) ocular symptoms, 3) oral symptoms, 4) parotid gland enlargement, 5) serological test results such as antibody to SSA (Ro) antigen (Anti-Ro/SSA), 6) antibody to SSB (La) antigen (Anti-La/SSB), 7) anti-nuclear antibody (ANA), 8) rheumatoid factor (RF); 9) diagnostic tests such as minor or labial salivary gland biopsy, 10) ocular staining test, 11) salivary flow test, and 12) S1 File on medications. For the thirteenth question, the reviewers assigned the patient's SD clinical

characteristics as positive, uncertain, or negative based on their responses and the guidelines (S1 Table in S1 File). The criteria for SD classification based on the characteristic findings included two categories 1) objective findings: histopathological, serological, ocular, and oral tests, and 2) subjective symptoms oral, ocular dryness, and parotid gland enlargement.

Domain experts in rheumatology (SH) and oral medicine/oral pathology (SZ) reviewed 20 patient records. They reviewed and finalized the guidelines based on their knowledge, experience and EHR findings. After the guidelines were finalized, two reviewers (2 registered nurses), were trained and calibrated using the same set of patients. In addition, these reviewers examined an additional 70 patient records in two rounds consisting of 20 and 50 records. Any disagreements regarding their findings were discussed with experts and resolved through consensus. After achieving an inter-rater agreement of 90%, the two calibrated reviewers examined and characterized/classified SD presentations for the remaining records independently. Once the reviews were completed, three research team members with clinical backgrounds (GFG, BC, TPT) reviewed the responses for correctness. Any concerns were discussed and resolved between the reviewers, experts and the research team through consensus and final SD classification for each patient was assigned.

### Manual review of EDR clinical notes for mention of Sjögren's Disease (SD) diagnosis

Guidelines were also developed to classify dental patients into positive, uncertain, or no mention of SD status based on EDR documentation using the criteria in Table 2. Two researchers reviewed clinical/progress notes, medical history, and medical consultation forms present in the EDR of the study cohort with a SD diagnostic code in their EHR using these guidelines. A third research team member (TPT) reviewed and discussed disagreements with the reviewers and finalized mention of SD in the EDR through consensus. Finally, for the overall study cohort, we determined how many of them received dental treatment/s before and after the SD index date.

### Data analysis

Data analysis and statistical testing were performed using the statistical analysis software SAS version 9.4 (SAS Institute, Inc., Cary, NC). Descriptive statistics including frequencies (N),

**Table 2. Guidelines to classify study cohort's Sjögren's Disease documentation in the electronic dental record.**

| Diagnosis | Criteria to classify SD patients' status as documented in EDR | | |
| --- | --- | --- | --- |
| | Positive (mention of confirmed SD) | Uncertain (mention of possible SD) | No mention of SD |
| Clinical notes and medical history form in EDR | • as a diagnosis in past medical history, clinical notes<br>• patient-reported records<br>• with diagnostic tests<br>• as a diagnosis<br>• medications (Rx for SD)<br>• by a rheumatologist<br>• dryness and signs | • diagnosis<br>• referral or evaluation by a specialist<br>• lip biopsy referral<br>• secondary SD<br>• dryness and no signs | No information of SD<br>• in medical history<br>• clinical notes<br>• no symptoms or findings |

Rx: medical prescription found in the EDR clinical note; SD: Sjögren's disease.

For the analysis, the date of first SD diagnostic code entered in the EHR is considered the index date and the patient age is calculated on this index date based on their date of birth entered in the EDR. Frequencies and percentages were calculated to describe the combinations of signs and symptoms among the three SD patient groups and prevalence of other autoimmune conditions, and medical conditions. Chi-square tests were performed for determining the association between the three SD characteristics/classifications within the EHR and other autoimmune and medical conditions among the study cohort.

means and standard deviations were calculated to summarize the demographics of the study population and compare the SD diagnosis determined from EHR data to the SD information recorded in the EDR data.

## Results

### Study population characteristics

Out of the 195,503 dental patients with a completed dental treatment visit between January 1, 2005, and December 31, 2020, at IUSD, 162,817 patients (83%) had EHR data in the INPC-R database (see Fig 1). Querying the patients' EHR for SD diagnostic codes (see Fig 1) yielded 441 distinct patients with at least one treatment or hospitalization visit and clinical notes. Based on the exclusion criteria, 64 patients were excluded resulting in a cohort of 377 EDR-EHR patient records to examine for SD clinical findings. Classification of SD clinical presentations through manual review of EHR resulted in 90 (24%) positive SD patients, 74 (20%) uncertain SD patients, and 213 (56%) negative SD patients (Table 3).

The mean age of the final study cohort (N = 377) was 54.31±14.35 standard deviation years. Females represented 91% of the study population. Eighty-four percent of this cohort were aged above 40 years (Table 3). A gradual increase in the proportion of positive and negative SD is observed with groups increasing in age but declining in ages 60 years and above. However, for the uncertain SD patients, most patients (30%) were in the category 60–69 years age group. The majority of the positive SD patients (32%) were diagnosed between 50 to 59 years of age. Most of the patients (71%) self-paid for their dental treatments. Of those with a positive SD diagnosis, 62% self-paid for their dental treatments.

### Frequency distribution of subjective symptoms and objective findings

The heatmap in Fig 2 represents the frequency distribution of subjective symptoms and objective findings, providing a comparison of positive, uncertain, and negative SD patient groups. Each rectangular cell in the matrix represents the total number of patients for the respective symptom or sign. The three symptoms and seven objective tests for signs are given as columns within the assigned SD status provided as rows. Hierarchical clustering of the variables among the SD classifications is provided as row dendrograms, and the symptoms and signs are given as column dendrograms. A cluster of similar data from the rows such as the negative and uncertain for SD are joined as a node. The positive SD cohort is a distinct cluster from the negative and uncertain SD cohorts. For instance, within symptoms, dry mouth and dry eyes are a cluster with higher frequency counts and within objective findings, salivary gland biopsy, salivary flow test and ocular staining test are a cluster with low frequency counts.

Table 4 shows the characteristics and predictive combinations of subjective symptoms and objective findings for the three SD patient groups. About 51% of the study cohort had symptoms of dry mouth and dry eyes and 17% had positive Anti-Ro/SSA antibodies and antinuclear antibody (ANA) test results. Dry mouth and dry eyes were accompanied with parotid gland enlargement in 6% of the cohort. For positive SD patients, 89% had both dry mouth and dry eyes and 77% had antibodies to Anti-Ro/SSA and ANA. About 10% had a report of positive focus score for salivary gland biopsy, one percent had a report of positive ocular staining, and one percent had reduced salivary flow rate. About 78% of patients within the uncertain SD group reported dry mouth and dry eyes and 62% had a positive RF finding. For patients within the negative SD group, 40% did not have symptoms or signs reported in their EHR data, and 26% reported dry mouth and dry eyes symptoms but did not have objective findings in their EHR (Table 4).

**Table 3. Sjögren's disease cohort demographics and by three clinical classifications.**

| | Patients with SD Diagnosis | | SD groups classified by clinical characteristics | | | | | |
| --- | --- | --- | --- | --- | --- | --- | --- | --- |
| | **Total** | | **Positive** | | **Uncertain** | | **Negative** | |
| | **N** | **(%)** | **N** | **(%)** | **N** | **(%)** | **N** | **(%)** |
| | **377** | **(100)** | **90** | **(23.9)** | **74** | **(19.6)** | **213** | **(56.5)** |
| Age group in years* | | | | | | | | |
| <20 | 2 | (0.5) | 0 | (0) | 0 | (0) | 2 | (0.9) |
| 20–29 | 19 | (5) | 4 | (4.4) | 5 | (6.8) | 10 | (4.7) |
| 30–39 | 41 | (10.9) | 8 | (8.9) | 10 | (13.5) | 23 | (10.8) |
| 40–49 | 75 | (19.9) | 17 | (18.9) | 18 | (24.3) | 40 | (18.8) |
| 50–59 | 94 | (24.9) | 29 | (32.2) | 11 | (14.9) | 54 | (25.4) |
| 60–69 | 94 | (24.9) | 25 | (27.8) | 22 | (29.7) | 47 | (22.1) |
| 70–79 | 41 | (10.9) | 6 | (6.7) | 7 | (9.5) | 28 | (13.1) |
| 80+ | 11 | (2.9) | 1 | (1.1) | 1 | (1.4) | 9 | (4.2) |
| Ethnicity | | | | | | | | |
| Hispanic or Latino | 25 | (6.6) | 7 | (7.8) | 8 | (10.8) | 10 | (4.7) |
| Not Hispanic or Latino | 317 | (84.1) | 79 | (87.8) | 61 | (82.4) | 177 | (83.1) |
| Other/unknown | 35 | (9.3) | 4 | (4.4) | 5 | (6.8) | 26 | (12.2) |
| Sex | | | | | | | | |
| Female | 343 | (91) | 86 | (95.6) | 69 | (93.2) | 188 | (88.3) |
| Male | 34 | (9) | 4 | (4.4) | 5 | (6.8) | 25 | (11.7) |
| Dental insurance | | | | | | | | |
| Government | 9 | (2.4) | 3 | (3.4) | 2 | (2.7) | 4 | (1.9) |
| Grant | 2 | (0.5) | 0 | (0) | 1 | (1.4) | 1 | (0.5) |
| Private | 99 | (26.4) | 31 | (34.8) | 16 | (21.6) | 52 | (24.5) |
| Self-Pay | 265 | (70.7) | 55 | (61.8) | 55 | (74.3) | 155 | (73.1) |
| Race | | | | | | | | |
| Asian/Pacific Islander | 6 | (1.6) | 1 | (1.1) | 1 | (1.4) | 4 | (1.9) |
| Black/African American | 95 | (25.2) | 32 | (35.6) | 11 | (14.9) | 52 | (24.4) |
| Multiracial | 3 | (0.8) | 1 | (1.1) | 2 | (2.7) | 0 | (0) |
| Other/Unknown | 8 | (2.1) | 2 | (2.2) | 2 | (2.7) | 4 | (1.9) |
| White | 265 | (70.3) | 54 | (60.0) | 58 | (78.4) | 153 | (71.8) |

*Patient age was calculated using the patient's date of birth and study index date. SD indicates Sjögren's disease; N indicates frequency counts; % indicates percentage for the demographic variables.

## Comorbidities including other autoimmune conditions in the study population

Among the 377 patients, 53% had at least one diagnosis of other autoimmune conditions in their EHR (Fig 3; S2 Table in S1 File). The most frequent conditions were rheumatoid arthritis (25%) followed by systemic lupus erythematosus (20%), and inflammatory arthropathy (12%). S2 Table in S1 File provides the percentages and p-values of related autoimmune comorbid conditions seen in this study population based on their classified group. Among positive SD patients, 33% had systemic lupus erythematosus and 30% had rheumatoid arthritis. Significant differences existed across the three SD groups for certain autoimmune conditions such as chronic lymphocytic thyroiditis (p = 0.027), psoriasis (p = 0.015), Raynaud's syndrome (p = 0.012), rheumatism (unspecified) and fibrositis (p = 0.034), rheumatoid arthritis (p = 0.005), systemic involvement of connective tissue (unspecified) (p = 0.001), and systematic lupus erythematosus (p<0.001) (S2 Table in S1 File).

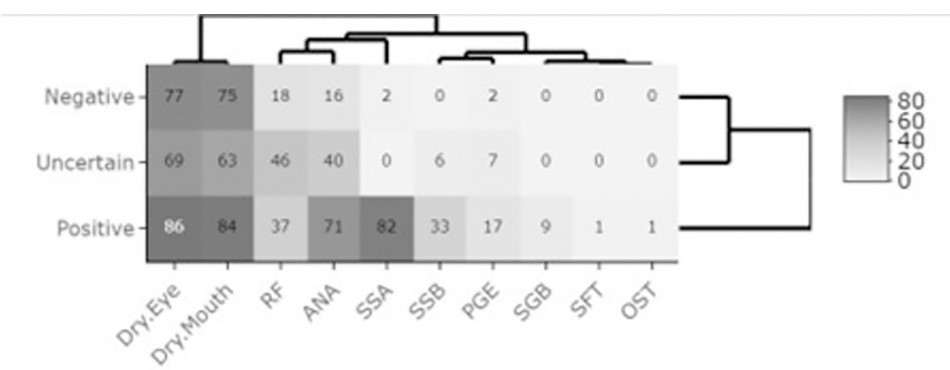

**Fig 2. Heat map visualization for signs and symptoms corresponding to SD classification.** RF–Rheumatoid factor; ANA–Anti-nuclear antibody; PGE–Parotid gland enlargement; SSB–antibody to SSB (La) antigen; OST–Ocular staining test; SGB–Salivary gland biopsy; SFT–Salivary flow test; Anti-Ro/SSA–antibody to SSA (Ro) antigen; DE–Dry eye; DM–Dry mouth.

Multiple comorbid conditions were also seen in 98% of the study cohort other than autoimmune conditions: 78% had pain in joints, and 72% had hypertension, followed by 63% with esophageal reflux (see Fig 4 and S3 Table in S1 File). In patients within the positive SD group, 74% reported pain in joints, followed by hypertension (72%), esophageal reflux (60%), depressive disorder (60%), anemia (54%) and osteoarthritis (42%). Fig 4 and S3 Table in S1 File illustrates frequently reported comorbid medical conditions in the EHR for this study cohort. Pulmonary hypertension (p = 0.034), osteoarthritis (p = 0.047), and hypercholesterolemia (p = 0.007) were significantly different at the 0.05 level across the three SD patient groups (S3 Table in S1 File).

## Documentation of Sjögren's disease in the electronic dental records

Reviewing clinical notes and medical histories of the study cohort's (N = 377) EDR revealed that 69% (N = 259) did not have any mention of SD (Table 5). Out of those without SD information in EDR, 50% (N = 45) belonged to the positive SD group, 64% (N = 47) to the uncertain SD group and 78% (N = 167) of patients to the negative SD group (Table 5). Sixteen percent of patients within the negative SD group based on our classification reported having SD during dental care, deeming positive SD diagnosis in EDR. In addition, 5% (11) of negative SD patient group in the EHR mentioned possible SD diagnosis and/or referral to rheumatologist or oral pathologist during dental care (see Table 5).

Of the 377 study patients, 118 (31%) had a mention of SD at any time during their dental visit in the EDR (Table 6). About 68% (N = 258) of these patients had a dental visit after their SD index date, with 21% (54) belonging to the positive SD group, 18% (N = 47) to the uncertain SD group and the remaining 61% (N = 157) to the negative SD group. Among patients in the SD negative group, 78% (N = 167) did not report their SD diagnosis during a dental visit (Table 6).

## Discussion

Through this study, we established a set of criteria for characterizing and classifying clinical findings of patients with a SD diagnosis in the EHR (Table 1). These criteria could be applicable for classifying SD patients using EHR data for research. Other major findings include variations in the EHR clinical findings contributing to the three SD patient groups (Fig 3, and Table 4), absence of an objective investigation of sicca symptoms for most study patients

**Table 4. Combination of Sjögren's disease symptoms and signs in the three groups by clinical characteristics.**

| SD Group | Symptoms | Signs | Count | (%) |
|---|---|---|---|---|
| Positive | DM+ DE | Anti-Ro/SSA+ANA | 20 | (22.2) |
| | DM + DE | Anti-Ro/SSA+ anti-La/SSB +RF+ANA | 14 | (15.6) |
| | DM + DE | Anti-Ro/SSA+RF+ANA | 11 | (12.2) |
| | DM + DE | Anti-Ro/SSA+ anti-La/SSB +ANA | 7 | (7.8) |
| | DM +DE +PGE | Anti-Ro/SSA+ANA | 6 | (6.7) |
| | DM + DE | Anti-Ro/SSA | 3 | (3.3) |
| | DM+ DE+ PGE | Anti-Ro/SSA+ anti-La/SSB +RF+ANA | 3 | (3.3) |
| | DM | Anti-Ro/SSA+ anti-La/SSB | 2 | (2.2) |
| | DM+ DE | SGB | 2 | (2.2) |
| | DM+ DE | ANA + SGB | 2 | (2.2) |
| | DM + DE | RF + SGB | 2 | (2.2) |
| | DM +DE+ PGE | Anti-Ro/SSA+ anti-La/SSB +ANA | 2 | (2.2) |
| | Dry Eye | Anti-Ro/SSA | 1 | (1.1) |
| | DE | Anti-Ro/SSA+ANA | 1 | (1.1) |
| | DE | Anti-Ro/SSA+RF | 1 | (1.1) |
| | DE | Anti-Ro/SSA+RF+ANA | 1 | (1.1) |
| | DE | Anti-Ro/SSA+ anti-La/SSB | 1 | (1.1) |
| | DE | Anti-Ro/SSA+ anti-La/SSB +RF+ANA | 1 | (1.1) |
| | DM | Anti-Ro/SSA | 1 | (1.1) |
| | DM | Anti-Ro/SSA+ anti-La/SSB + ANA+ OST | 1 | (1.1) |
| | DM + DE | Anti-Ro/SSA+RF +ANA + SGB | 1 | (1.1) |
| | DM + DE | Anti-Ro/SSA+ anti-La/SSB | 1 | (1.1) |
| | DM + DE +PGE | SFT | 1 | (1.1) |
| | DM + DE +PGE | RF + SGB | 1 | (1.1) |
| | DM +DE +PGE | Anti-Ro/SSA | 1 | (1.1) |
| | DM+ DE +PGE | Anti-Ro/SSA+RF | 1 | (1.1) |
| | DM +DE +PGE | Anti-Ro/SSA+RF+ANA | 1 | (1.1) |
| | DM +DE +PGE | Anti-Ro/SSA+ anti-La/SSB + SGB | 1 | (1.1) |
| Uncertain | DM + DE | RF | 22 | (29.7) |
| | DM + DE | ANA | 14 | (18.9) |
| | DM + DE | RF+ANA | 11 | (14.9) |
| | DE | RF | 6 | (8.1) |
| | DM + DE | anti-La/SSB | 4 | (5.4) |
| | DM | ANA | 3 | (4.1) |
| | DM + DE + PGE | ANA | 3 | (4.1) |
| | DE | ANA | 2 | (2.7) |
| | DE | RF+ANA | 2 | (2.7) |
| | DM + DE + PGE | RF + ANA | 2 | (2.7) |
| | DE | anti-La/SSB +ANA | 1 | (1.4) |
| | DM | RF | 1 | (1.4) |
| | DM | RF+ANA | 1 | (1.4) |
| | DM + DE + PGE | RF | 1 | (1.4) |
| | DM + DE +PGE | anti-La/SSB + ANA | 1 | (1.4) |

(*Continued*)

**Table 4.** (Continued)

| SD Group | Symptoms | Signs | Count | (%) |
|---|---|---|---|---|
| Negative | No symptoms | Objective tests unavailable | 86 | (40.4) |
| | DM + DE | Objective tests unavailable | 55 | (25.8) |
| | DE | Objective tests unavailable | 21 | (9.9) |
| | DM | Objective tests unavailable | 19 | (8.9) |
| | No symptoms | RF | 14 | (6.6) |
| | No symptoms | ANA | 9 | (4.2) |
| | No symptoms | RF + ANA | 4 | (1.9) |
| | No symptoms | Anti-Ro/SSA+ANA | 2 | (0.9) |
| | PGE | Objective tests unavailable | 2 | (0.9) |
| | DM + DE | *ANA | 1 | (0.5) |

*Lower titer values of ANA compared to the cut-off level in our criteria along with symptoms. SD–Sjögren's disease; DM–mouth; DE–Dry eye; PGE–Parotid gland enlargement; Anti-SSA (Ro)–antibody to SSA (Ro) antigen; Anti-SSB (La)—antibody to SSB (La) antigen; RF–Rheumatoid factor; ANA–Anti-nuclear antibody; OST—Ocular staining test; SGB–Salivary gland biopsy; SFT–salivary flow test

(Table 4), high prevalence of certain comorbidities including other autoimmune and medical conditions (Figs 3 and 4), and gaps in the documentation of SD diagnosis during dental visits (Tables 5 and 6). To the best of our knowledge, this study is the first to characterize the EHR clinical findings of patients with a SD diagnosis and documentation of SD in their EDR.

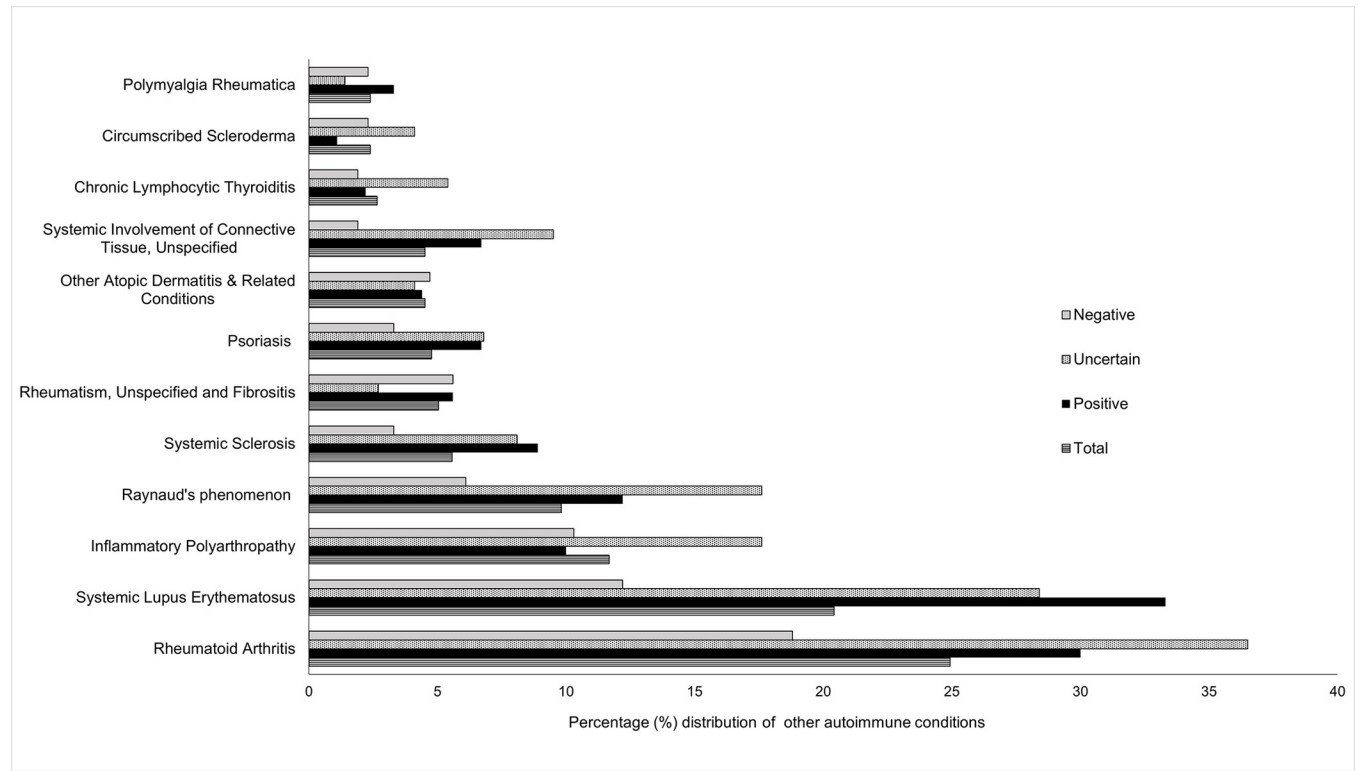

**Fig 3. Other autoimmune conditions in the study cohort and by three Sjögren's disease (SD) groups.** Other autoimmune conditions such as rheumatoid arthritis, systemic lupus erythematous, inflammatory polyarthropathy represent more than 10% among SD patients. Total: study cohort; positive, uncertain, negative represent the three SD groups by clinical characteristics.

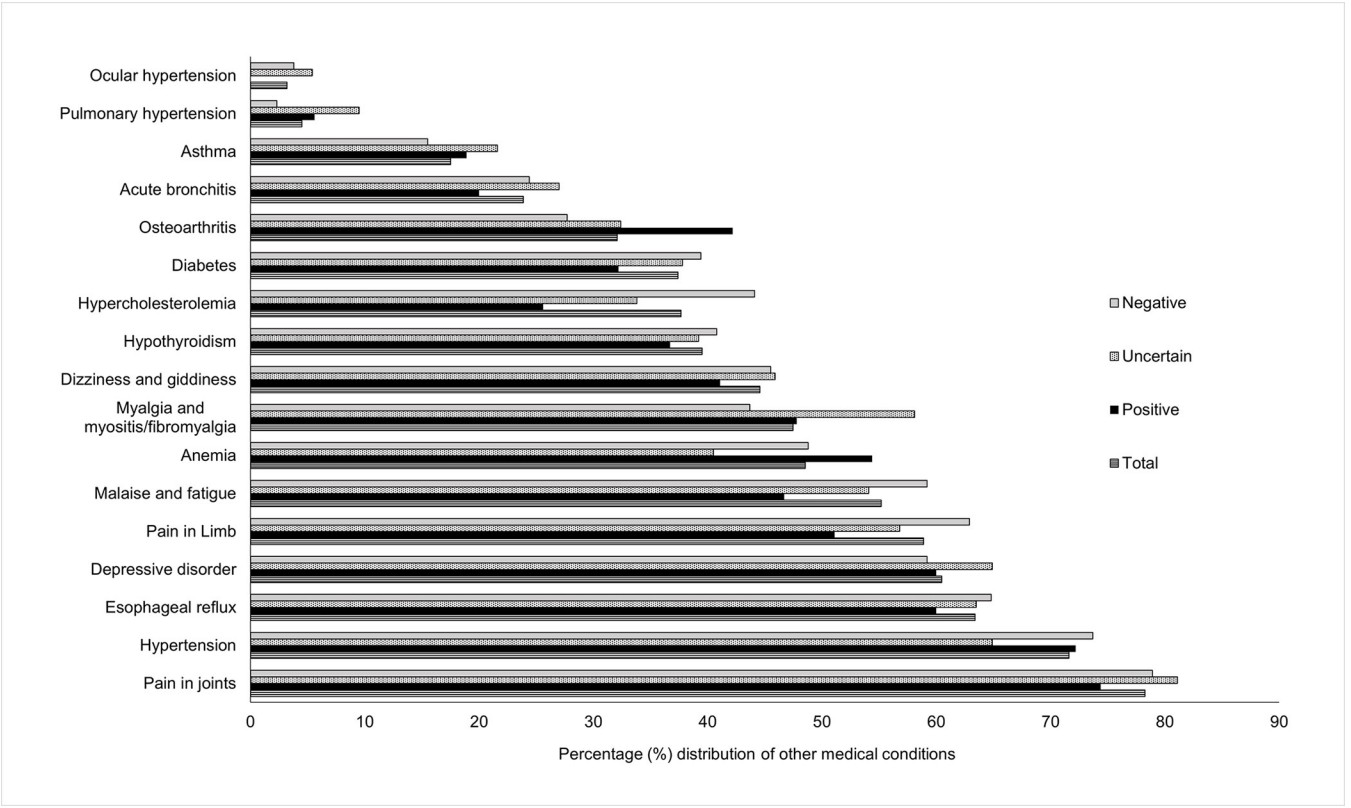

**Fig 4. Other medical conditions present in the study cohort and by the three Sjögren's disease groups.** SD: Sjögren's disease total: study cohort; positive, uncertain, negative represent the three SD groups within the study cohort by clinical characteristics. Medical conditions such as pain in joints, hypertension, esophageal reflux, depressive disorder, pain in limb, malaise and fatigue is present in ≥50% SD cohort.

The characteristics of this study population as evidenced through the results are consistent with previously reported studies based on the AECG criteria [9, 26, 44]. This study cohort had 91% females consistent with previous reports of female to male predilection of 9:1 [1, 3, 9, 14, 45–51]. In addition, the average age for females was 55 years and it is consistent with reporting of SD over the age of 50 and around menopause [1, 3, 9, 45, 48–50]. Additionally, previous reports and this study showed that physicians use mostly serology test results to diagnose SD [9, 26, 30]. As a result, the classifications used to confirm a person's SD diagnosis to enroll in clinical research cannot be used to generate SD cohorts using EHR or EDR data. The criteria

**Table 5. Documentation of Sjögren's disease in electronic dental records for the three groups.**

| SD study cohort classified based on EHR clinical findings | SD study cohort classified based on SD documentation in the electronic dental record | | | | |
|---|---|---|---|---|---|
| | Positive | | Uncertain | | No mention of SD/ without SD | Total | (%) |
| | N | (%) | N | (%) | N | % | N | (%) |
| Positive | 41 | (45.6) | 4 | (4.4) | 45 | (50.0) | 90 | (23.9) |
| Uncertain | 26 | (35.1) | 1 | (1.4) | 47 | (63.5) | 74 | (19.6) |
| Negative | 35 | (16.4) | 11 | (5.2) | 167 | (78.4) | 213 | (56.5) |
| Total | 102 | (27.1) | 16 | (4.2) | 259 | (68.7) | 377 | (100) |

SD: Sjögren's Disease; EHR: electronic health record; EDR: electronic dental record.

**Table 6. Documentation of Sjögren's disease in electronic dental record before and after the index date.**

| SD study cohort classified based on EHR clinical findings | SD study cohort's documentation of SD in the EDR | Dental treatment after the Sjögren's disease diagnosis index date in the EHR | | | | | |
| --- | --- | --- | --- | --- | --- | --- | --- |
| | | Yes | | No | | Total | |
| | | N | (%) | N | (%) | N | (%) |
| Positive | Positive | 34 | (82.9) | 7 | (17.1) | 41 | (45.6) |
| | Uncertain | 2 | (50.0) | 2 | (50.0) | 4 | (4.4) |
| | No mention of SD | 18 | (40.0) | 27 | (60.0) | 45 | (50.0) |
| | **Total** | **54** | **(60.0)** | **36** | **(40.0)** | **90** | **(100)** |
| Uncertain | Positive | 24 | (92.3) | 2 | (7.7) | 26 | (35.1) |
| | Uncertain | 0 | (0) | 1 | (100) | 1 | (1.4) |
| | No mention of SD | 23 | (48.9) | 24 | (51.1) | 47 | (63.5) |
| | **Total** | **47** | **(63.5)** | **27** | **(36.5)** | **74** | **(100)** |
| Negative | Positive | 32 | (91.4) | 3 | (8.6) | 35 | (16.4) |
| | Uncertain | 10 | (90.9) | 1 | (9.1) | 11 | (5.2) |
| | No mention of SD | 115 | (68.9) | 52 | (31.1) | 167 | (78.4) |
| | **Total** | **157** | **(73.7)** | **56** | **(26.3)** | **213** | **(100)** |
| **Total** | | **258** | **(68.4)** | **119** | **(31.6)** | **377** | **(100)** |

SD: Sjögren's Disease; EHR: electronic health record; EDR: electronic dental record.

developed in this study based on clinical experts' recommendations could be used to generate SD cohorts with different clinical presentations from the EHR data for longitudinal studies.

## Variations in findings contributing to SD diagnosis

The nonspecific and heterogenous presentation of SD symptoms and findings as reported in previous studies was also evident in this study cohort [10, 15, 26, 30, 52–54]. Although, we used the defined criteria to classify patients as positive, uncertain, or negative, all patients have a diagnostic code of SD irrespective of multiple clinical presentations. As reported in recent studies, these heterogenous symptoms could be because definite symptoms and findings indicative of SD may become obvious only later in the disease process [18, 55]. For instance, the negative SD group patients had less severe findings when compared to positive and uncertain SD groups indicating variations in disease severity in this cohort diagnosed in community practices. In addition, since the primary management of SD continues to be symptomatic, healthcare providers may prefer to diagnose SD even in the absence of definitive findings so that patients benefit from symptomatic management and relief [56]. Even though it is quintessential to educate clinicians on the possibility of misdiagnosing SD patients in clinical setting, it is likely that it will continue because of the complexity of the disease. However, for research purposes, a combination of symptoms and objective test results are used to define a homogenous population with SD for clinical studies enrollment. As a result, most clinical study results may not be generalizable to patients diagnosed in community practice settings due to the different criteria used to diagnose to capture this condition's varying clinical presentations during patient care. This study extends previous research by quantifying the different combinations of symptoms and objective findings used to diagnose SD in clinical practices. These findings could be the first step to phenotype different presentations of SD observed in clinical settings using EHR data.

## Absence of objective investigation of sicca symptoms for most study patients

Although dry mouth and dry eyes (sicca symptoms) are primary symptoms for SD, only two cases reported using salivary flow rate and ocular staining test to diagnose SD in this study

cohort. However, minor salivary gland biopsy results indicated SD was present for 9 patients (2%). Another important finding is the absence of sicca symptoms and any test results indicating SD for 86 patients (40%) in the SD negative patient group according to this study guideline (Table 4). These results suggest that physicians may apply the presence of disease activity other than sicca symptoms and test results to assign an SD diagnosis. As mentioned above, a possible reason for using such wide diagnostic criteria could be to detect accurately all the people with this condition.

Moreover, serology test results such as anti-Ro/SSA, RF and ANA were predominantly used to diagnose SD. Physicians' easy access to order serology tests could be a primary reason for using these tests more frequently than other tests such as salivary flow test, ocular staining score and minor salivary gland biopsy. Previous studies have also reported using serology test results to diagnose SD [26]. Prevalence of dryness symptoms along with anti-Ro/SSA and ANA were increased compared to other serological tests in SD positive patient group. Similar to the American College of Rheumatology (ACR) criteria using combined RF and ANA (1:320), our study indicates use of combined anti-Ro/SSA and ANA (1:160) [21, 57]. Presence of only positive anti-La/SSB with negative anti-Ro/SSA does not conform to phenotypic characteristic features of SD patients [6, 44]. The presence of positive RF and ANA results in this study cohort could be attributed to not distinguishing between patients with primary and secondary SD. Also, the need to refer patients to other specialists such as dentists, oral surgeons, oral pathologists, ophthalmologists, or otolaryngologists although primary management continues to be symptomatic may prevent physicians and patients from seeking additional test results. Nevertheless, given that sicca symptoms could have adverse effects on SD patients' oral and ophthalmologic health resulting in tooth loss and corneal scarring, establishing multidisciplinary care for these patients is crucial.

## Display of other concomitant autoimmune diseases and medical conditions

About 53% of the study cohort had other autoimmune conditions that could be considered as secondary SD. The EHR data display rheumatoid arthritis, systemic lupus erythematosus (SLE), and Raynaud's phenomenon affecting at least 15–30% of these patients. Anti-Ro/SSA and ANA are common among other autoimmune diseases, and it is required to delineate the conditions based on antibody titer values and objective tests. Pain in joints, pain in the limb, hypertension, fibromyalgia, malaise and fatigue, esophageal reflux, and depressive disorders are the comorbidities affecting more than half of this patient population in the study cohort. These findings are consistent with the comorbidities reported in previous studies [8, 18, 29, 58–60]. These comorbidities, except for pulmonary hypertension, osteoarthritis, and hypercholesterolemia, have similar distributions among the three classified groups in this cohort. Given the high prevalence and similar distribution of most comorbidities, screening adults with these conditions and investigating related biomarkers may identify SD before the established clinical manifestation of SD.

## Gaps in the documentation of SD diagnosis during dental visits

Previous research investigating the oral health of patients with SD has been primarily prospective clinical studies where SD is confirmed according to established criteria for clinical studies. To the best of our knowledge, this retrospective study is the first to determine the SD diagnosis of dental patients by accessing SD diagnosis from EHR data. Unfortunately, only 31% (n = 118) of these patients with an EHR SD diagnosis had documentation of a SD diagnosis at any time during their dental visits. This finding is not surprising given the discrepancies between EHR and EDR data on dental patients' medical conditions, such as diabetes [61],

cardiovascular diseases [62], and medication histories [63]. While medical providers must record diagnosis to support their treatment plan, dental providers are not required to record diagnosis. This difference may contribute to the significant differences in the recording of medical and dental conditions in EDR. Nevertheless, as best practices, it is necessary that dental providers maintain good documentation of their information gathering, assessment and management of patients.

Regardless, given that the adverse impact of hyposalivation due to SD on a person's oral health is a well-known fact, missing this diagnosis during dental care prevents this population from receiving additional preventive care. The results also indicate the potential gaps in care due to the absence of adequate care coordination between different healthcare specialties involved in caring for SD patients. Patients diagnosed with SD require multidisciplinary care that includes, at a minimum, physicians, rheumatologists, dental professionals, and ophthalmologists. Therefore, appropriate systems need to be in place to facilitate information sharing and communication between different healthcare providers to provide optimum care not only to SD patients but also for individuals who are affected with conditions that affect all systems of the human body.

## Limitations

As with any study, this study also has limitations. First, we could not confirm availability of diagnostic tests including serology tests for all patients from other sources such as laboratory records not integrated into the EHR data. Therefore, we classified patients who have no available diagnostic tests in the EHR, but with presence of symptoms as SD negative group based on our criteria. Some of these patients may have a diagnostic test in a different healthcare system not captured in our database, or they may not have an available laboratory test. Second, this study is based on linking EDR data from one institution with the EHR data in the statewide HIE. Therefore, the generalizability of the diagnostic criteria used to define SD clinical presentations needs to be investigated with EHR data sets from other community practice settings. Nevertheless, these criteria can be deemed generalizable given the clinical presentations and demographics of this SD study cohort are consistent with that reported in previous studies including the ones that used the established AECG criteria. Third, we did not distinguish the clinical characteristics between primary and secondary SD patients although 53% of this cohort could be considered as secondary SD due to presence of other autoimmune conditions. Future work should investigate the similarities and differences in the clinical characteristics between primary and second SD patient groups.

## Conclusion

This study developed a set of criteria that characterizes the different clinical presentations of dental patients with SD diagnosis in community practices using EHR data. The results emphasize the heterogenous SD clinical characteristics presented by patients during clinical care. Also, there is a need to investigate disease progression and treatment outcomes and further research to diagnose SD early in community practices. Although diagnostic codes were used to identify SD patients, it is also necessary to review the clinical notes in electronic health records and access the patients' laboratory reports for research studies as well for patient care. The developed criteria could be applied for future longitudinal studies using EHR data of SD patients based on different clinical characteristics and severity of the disease. Further studies are needed to determine the prognostic factors in the early onset of the disease and the effectiveness of preventive caries management among SD patients. Additionally, there is also the need for screening and awareness of SD disease and its symptoms should be promulgated

through educational programs targeting women. Finally, this study reveals the importance of linking medical and dental electronic records for providing better oral care for SD patients in community dental clinical settings.

## Supporting information

**S1 File.**
(DOCX)

## Acknowledgments

We thank Ms. Lucy Bickett, M.J, Ms. Jessica Esch, M.A, Ed.D candidate for reviewing, proof-reading, and editing the manuscript. We acknowledge Ms. Sue Mohning and Ms. Taffy French for reviewing clinical notes in EHR, Harsha Bandaru for assisting with reviewing results and Ms. April Currier, Mr. Jarod Baker, and Mr. Anthony Vilhauer for the successful management of this project.

## Author Contributions

**Conceptualization:** Thankam P. Thyvalikakath.

**Data curation:** Grace Gomez Felix Gomez, Jay S. Patel, Mei Wang, Anushri Singh Rajapuri, Lauren R. Lembcke, Divya Rajendran, Jonas C. Smith, Biju Cheriyan, Thankam P. Thyvalikakath.

**Formal analysis:** LaKeisha J. Boyd, George J. Eckert.

**Funding acquisition:** Thankam P. Thyvalikakath.

**Investigation:** Grace Gomez Felix Gomez, Steven T. Hugenberg, Susan Zunt, George J. Eckert, Shaun J. Grannis, Mythily Srinivasan, Domenick T. Zero.

**Methodology:** Grace Gomez Felix Gomez, Steven T. Hugenberg, Susan Zunt, Jay S. Patel, Mei Wang, Lauren R. Lembcke, Biju Cheriyan, George J. Eckert, Shaun J. Grannis, Domenick T. Zero, Thankam P. Thyvalikakath.

**Project administration:** Thankam P. Thyvalikakath.

**Resources:** Thankam P. Thyvalikakath.

**Supervision:** Thankam P. Thyvalikakath.

**Validation:** Grace Gomez Felix Gomez, Steven T. Hugenberg, Susan Zunt, Biju Cheriyan, Thankam P. Thyvalikakath.

**Writing – original draft:** Grace Gomez Felix Gomez.

**Writing – review & editing:** Grace Gomez Felix Gomez, Steven T. Hugenberg, Susan Zunt, LaKeisha J. Boyd, George J. Eckert, Shaun J. Grannis, Mythily Srinivasan, Domenick T. Zero, Thankam P. Thyvalikakath.

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
