## [Decision Letter · Decision Letter 0]

27 Mar 2023

PONE-D-22-35694Characterizing Clinical Findings of Sjögren’s Disease Patients in Community Practices Using Matched Electronic Dental-Health Record Data.PLOS ONE

Dear Dr. Thyvalikakath,

Thank you for submitting your manuscript to PLOS ONE. After careful consideration, we feel that it has merit but does not fully meet PLOS ONE’s publication criteria as it currently stands. Therefore, we invite you to submit a revised version of the manuscript that addresses the points raised during the review process.

We look forward to receiving your revised manuscript.

Kind regards,

Boyen Huang, DDS, MHA, PhD

Academic Editor

PLOS ONE

Journal Requirements:

"This research was supported by the National Institutes of Health grants R21 DE027786-02 and 1R56DE029195-01."

"Corresponding and senior author (TPT) received federal support for a research project through National Institutes of Health/National Institute of Dental and Craniofacial Research (NIH/NIDCR) grants R21 DE027786-02 and 1R56DE029195-01. The project is to assess oral health and dental treatment outcomes among Sjögren's Syndrome patients using linked dental and health record data. The submitted manuscript is a part of the funded research project. The website link for the funding agency is https://www.nidcr.nih.gov/grants-funding. The funders had no role in study design, data collection and analysis, decision to publish, or preparation of the manuscript."

4. Thank you for stating the following in your Competing Interests section: "NO authors have competing interests."

5. We noted in your submission details that a portion of your manuscript may have been presented or published elsewhere:

"No. The findings from this work were not submitted to other journals. Data from this work have been presented at conferences and research meetings."

Please clarify whether this conference proceeding was peer-reviewed and formally published. If this work was previously peer-reviewed and published, in the cover letter please provide the reason that this work does not constitute dual publication and should be included in the current manuscript.

6. We note that you have indicated that data from this study are available upon request. PLOS only allows data to be available upon request if there are legal or ethical restrictions on sharing data publicly. For more information on unacceptable data access restrictions, please see http://journals.plos.org/plosone/s/data-availability#loc-unacceptable-data-access-restrictions. 

Reviewers' comments:

Reviewer's Responses to Questions

**Comments to the Author**

1. Is the manuscript technically sound, and do the data support the conclusions?

Reviewer #1: No

Reviewer #2: Yes

Reviewer #3: Yes

Reviewer #4: No

2. Has the statistical analysis been performed appropriately and rigorously? 

Reviewer #1: I Don't Know

Reviewer #2: Yes

Reviewer #3: Yes

Reviewer #4: Yes

3. Have the authors made all data underlying the findings in their manuscript fully available?

Reviewer #1: Yes

Reviewer #2: Yes

Reviewer #3: Yes

Reviewer #4: No

4. Is the manuscript presented in an intelligible fashion and written in standard English?

Reviewer #1: Yes

Reviewer #2: Yes

Reviewer #3: Yes

Reviewer #4: Yes

5. Review Comments to the Author

Reviewer #1: Characterizing Sjogren´s disease patients using electronic health and dental records

The authors end up with concluding that there is a need for establishing criteria to diagnose SD early in community practices. And the study criteria can be used to generate SD cohorts for longitudinal studies.

I can agree on the first statement, but it is a mystery to me how the study criteria developed here can contribute to this?

Introduction

The author intents to develop a paradigm or a screening tool for patients suspect of Mb Sjogren, enforcing the flow of potential patients to specialist care by linking two databases EDR and EHR.

Line 83-89

It is important to notice that Sjogren’s disease is a systemic connective disease not only relying on dryness of mouth and eyes. People have these subjective symptoms for many reasons together with constitutional symptoms without have Sjogren. And although diagnostic criteria are developed, the diagnosis rely on pattern recognition of rheumatologist. The AECG 2002 diagnostic criteria was developed to ensure so homogeneous cohort for research as possible. And internal medical doctors are aware of that Sjogren’s patients are different. Some have very heavy manifestations of the disease including peripheral neuropathy, cutaneous, pulmonary, and renal disease.

In the introduction 4 aims was set-up – have the authors answered them?

Line 113-114

Can the authors comment on the fact that these data bases not are performed as a research database, but for administrative registration and economic properties. Is there a possibility for bias of here?

Line 147 and 250

Eighty-three percent was successfully matched – what about the rest 17%? - Have you any information of the group? Could there be at selection bias here?

Line 164-169

What about smoking status? What about medical treatment? Diuretics? Anti-hypertensive? Anti-depressive? All factors that have a significant impact on the subjective symptoms but also the objective function tests.

Line 175-177 and table 1:

A clear overview of the criteria for categorizing the possible patients complaining of dry mouth or dry eyes into three groups.

Why have you chosen to register the occurrence of Ig M RF? And ANA The occurrence of rheumatoid factor and ANA do not qualify for diagnosis. Noise? See later.

Line 216

Heroic efforts were performed to train the reviewers.

Line 251 and fig 2

441 patients were picked out and 64 patients were excluded resulting in a cohort of 377. 23.9 % positive, 19.6% uncertain and 56.6% negative.

The occurrence of Ig M RF and ANA are the main two laboratory findings that qualify for the patients to be placed into the group of uncertain. Noise?

Line 329

The difference is comorbidity – Is it real or is a consequence of better health care among patients with inflammatory joint or connective disease?

Table 5

The most interesting table. Here we see the results of comparison of the two databases.

Comments of the conformity or lack thereof?

And can you speculate if one of the two databases comes closer to the truth? Or are you comparing to databases with great uncertainties?

Line 360

It is listed that these criteria could be applicable for classifying SS patients using EHR data for research. It is an assertion that cannot stand alone. For me it is not obviously according to the table 5.

S 373-374

This is the hot spot – Serology test can not be used to diagnose Sjogren or to screen for Sjogren.

Line 388-398 – and table 5 and line 403-405

Dry mouth and dry eyes are symptoms that can be treated symptomatically. But there are many reasons for these symptoms at not necessary a result of an ongoing autoimmune process. I am not familiar with your health care system – could there be an economic incitement to give the diagnoses to at patient. Repay for the bill from the dentist or access to be enrolled into clinical randomized trials for new treatments?

Line 403-406 and table 4

The authors are pointing to a very weak point in the daily clinic which are reflected in the records – misdiagnosis or misclassification. Could on of the point in this publication be that there is a urgent need for education.

Line 491-492

It is not clear for me how the criteria developed through this study can be applicable in future research that involves early diagnosis. What are the advantages of these versus the EACG criteria?

Reviewer #2: This robust study describes the development criteria for characterizing and classifying findings in electronic health and dental records for Sjogren's Disease patients. This was well written and there are just a few minor comments to consider below.

Abstract

- As the first study of its kind, the objective misses some opportunity to describe the fullness of the actual study - later in the paper there are 4 aims outlined but it is only written as "characterized" in the abstract.

Methods

- Consider including a brief explanation as to why those conditions were excluded in the study (line 159-164).

Conclusion

- Recommendations are stated but I am curious to how this work could be implemented in the EHR/EDRs at IUSD and beyond?

Reviewer #3: This is a retrospective study using electronic dental record (EDR) and electronic health record (EHR) data to improve the delayed diagnosis of the patients with Sjögren’s disease (SD) for practicing clinicians. It is interesting because it is unique and highly original, and the research methods and interpretation of the results were conducted without any problems.

However, in order to disseminate our diagnostic criteria widely and clinically, it is necessary to describe in more detail the results compared with the already established standard criteria of SD. I am afraid that the content of this text is insufficient. Please reconsider this point and resubmit it.

Reviewer #4: Thank you for this manuscript. You used a lot of data to find more clearity in the SD diagnosis mainly in dental health records. You do present a lot of numbers and results however the conclusions based on these numbers are somewhat vague to me. What do you conclude? And what are the results concerning your aims? For me it reads like a relatively large part of the patients most likely do not have SD (as they also have rheumatiod artritis or lupus). And what does the EDR data add to the EHR data. It is not clear for me from the results and discussion.

To me it feels like the results are not as positive as the authors hoped for. And therefore the results are kept a bit vague. Even if the result is not what you hoped for you should report this clearly. A 'negative' result is also a result.

6. PLOS authors have the option to publish the peer review history of their article (what does this mean?). If published, this will include your full peer review and any attached files.

Reviewer #1: No

Reviewer #2: **Yes: **Karmen S Williams

Reviewer #3: No

Reviewer #4: No

---

## [Author Response · Author response to Decision Letter 0]

15 Jun 2023

RESPONSE TO REVIEWERS

We thank reviewers for their time to review our manuscript. Comments from reviewers are in normal text and responses are italicized.

1. Comment: The authors end up with concluding that there is a need for establishing criteria to diagnose SD early in community practices. And the study criteria can be used to generate SD cohorts for longitudinal studies. I can agree on the first statement, but it is a mystery to me how the study criteria developed here can contribute to this?

Response: Thanks for the comments regarding establishing criteria to diagnose Sjögren’s disease (SD) early in community practices. We acknowledge the two statements are separate conclusions and not related to each other. This study focused on developing guidelines and carefully characterizing the different clinical presentations of SD in community settings using electronic health record data from multiple healthcare systems and practices. We conclude the results of this study will help other researchers generate study cohorts using EHR data to investigate disease progression and treatment outcomes. The study results also highlight the need for future research to investigate tests/criteria to detect SD early in community practice settings. To clarify these points, we have made the following changes in the abstract and the conclusion sections of the paper that emphasizes that the criteria can be used to generate cohorts using EHR data and not linked EDR-EHR data.

Revised [Lines 59-63]: This study of SD patients diagnosed in community practices characterized three different SD clinical presentations, which can be used to generate SD study cohorts for longitudinal studies using EHR data. The results emphasize the heterogenous SD clinical presentations and the need for further research to diagnose SD early in community practice settings where most people seek care.

Revised [Lines 489-494]: This study developed a set of criteria that characterizes the different clinical presentations of dental patients with SD diagnosis in community practices using EHR data. The results emphasize the heterogenous SD clinical characteristics presented by patients during clinical care,the need to investigate disease progression and treatment outcomes. and further research to diagnose SD early in community practices.

2. Comment: Line 83-89

It is important to notice that Sjogren’s disease is a systemic connective disease not only relying on dryness of mouth and eyes. People have these subjective symptoms for many reasons together with constitutional symptoms without have Sjogren. And although diagnostic criteria are developed, the diagnosis rely on pattern recognition of rheumatologist. The AECG 2002 diagnostic criteria was developed to ensure so homogeneous cohort for research as possible. And internal medical doctors are aware of that Sjogren’s patients are different. Some have very heavy manifestations of the disease including peripheral neuropathy, cutaneous, pulmonary, and renal disease.

In the introduction 4 aims was set-up – have the authors answered them?

Response: 

We agree with your observations. As you stated, physicians diagnose SD patients based on non-specific symptoms and other systemic involvement. These are not listed in the current classification criteria or in our study criteria which lead such cases to be classified as uncertain or negative. There is a need to establish criteria that can be applied in community practice settings to diagnose SD.

Yes, we have answered all four aims in the results as highlighted in the paper in lines 177 to 234, (including tables 1 and 2) in the methods, lines 291 to 302 including table 4 in the results and lines 363 to 369 in the discussion sections.

For aim 1, manual review guidelines were developed to classify Sjögren’s disease patients based on their clinical characteristics using electronic health record (EHR) data. Tables 1 & 2 present the manual review guidelines used in the methods. Table 3 provides the cohort characteristics by three classifications in the results. 

For aim 2, SD patients’ symptoms and signs combinations for three groups with their counts and percentages are provided in table 4.

For aim 3, Figures 3 and 4 provide the comorbid and autoimmune conditions of SD patients, and their associations are described in lines 306 to 331 as well in the supplemental tables (S2 and S3).

For aim 4, documentation of Sjögren’s disease in electronic dental records for the three groups are illustrated in table 5.

From Line 363 to 370 in the discussion, findings from all 4 aims are provided indicating all 4 aims have been completed. Please see the text below.

“Through this study, we established a set of criteria for characterizing and classifying clinical findings of patients with a SD diagnosis in the EHR (Table 1). These criteria could be applicable for classifying SS patients using EHR data for research. Other major findings include variations in the EHR clinical findings contributing to the three SD patient groups (Fig 3, and Table 4), absence of an objective investigation of sicca symptoms for most study patients (Table 4), high prevalence of certain comorbidities including other autoimmune and medical conditions (Figs 3 and 4.), and gaps in the documentation of SD diagnosis during dental visits (Tables 5 and 6).” 

3. Comment: Line 113-114

Can the authors comment on the fact that these data bases not are performed as a research database, but for administrative registration and economic properties. Is there a possibility for bias of here?

Response: 

The electronic dental record (EDR) data and the electronic health record (EHR) data include patient care information gathered during their dental and medical care visits and not administrative data. They include demographics, medical conditions, treatments received, laboratory and other diagnostic test results, medications, and clinical notes (https://www.ihie.org/). We agree that these data are not derived from prospective research studies. However, the data is derived from real-world clinical settings, enabling real-world evidence to assess treatment outcomes and disease progression. Also, in contrast to claims data, EDR and EHR data allow the study of uninsured populations’ disease and treatment outcomes. Therefore, the possibility of bias is less compared to studying claims data that include information only from insured populations.

4. Comment: Line 147 and 250

Eighty-three percent was successfully matched – what about the rest 17%? - Have you any information of the group? Could there be at selection bias here?

Response:

We are grateful for the suggestion to investigate the 17% unmatched group and will undertake it in our future research. It is possible this group may include patients coming from regions in states bordering Indiana such as Illinois and Kentucky because the Indiana University School of Dentistry is a major referral center for advanced dental care. These patients’ healthcare systems may be located outside of Indiana with less chance of contributing patient care information to the Indiana health information exchange (IHIE). Finally, this group may include patients who may be part of the small number of Indiana healthcare systems that are not yet participating in the IHIE. However, these reasons do not contribute to a selection bias in this study. 

As described in the methods section (lines 145 to 149), the Indiana Network for Patient Care-Research (INPC-R) database is maintained by the Indiana Health Information Exchange (IHIE), a mature state-wide health information exchange. INPC-R contains more than 20 million patient record data from over 123 hospitals and 19,000 practices in Indiana. Therefore, 83% match is considered significantly higher compared to matching with electronic health record system from one health care system. Anecdotal reports suggest less matched record rates ranging from 15%-50% when matching electronic dental record data with patients’ medical records present in one healthcare system. 

5. Comment: Line 164-169

What about smoking status? What about medical treatment? Diuretics? Anti-hypertensive? Anti-depressive? All factors that have a significant impact on the subjective symptoms but also the objective function tests.

Response:

We agree about studying smoking status, medical treatments, and medications. However, this study is a descriptive study characterizing the clinical presentations of dental patients with a diagnosis of SD using EHR data. Careful curation of clinical findings is critical before they can be analyzed, and therefore, this process is challenging and time-consuming. We also believe highlighting the different clinical presentations of SD is critical to stimulate further research to diagnose SD early in clinical settings. Therefore, in the next ongoing research, we are retrieving their treatments and medication history and classifying them into drug classes to assess dental treatment outcomes. We are also retrieving smoking status, which until recently, was recorded in the free text format within EHR and EDR, and require additional text mining approaches to retrieve this information. We plan to publish this information as a separate paper.

6. Comment: Line 175-177 and table 1: 

A clear overview of the criteria for categorizing the possible patients complaining of dry mouth or dry eyes into three groups.

Why have you chosen to register the occurrence of Ig M RF? And ANA The occurrence of rheumatoid factor and ANA do not qualify for diagnosis. Noise? See later.

Response: Using rheumatoid factor and ANA for SD diagnosis cannot be considered noise based on existing guidelines. The American College of Rheumatology and European League Against Rheumatism (EULAR) have determined ANA and RF as part of the classification criteria for SD (Shiboski et al, 2012). In addition, a previous US study (Maciel 2017) using EHR data reported antinuclear antibodies (ANA), anti-SSA, rheumatoid factor (RF) and anti-SSB as the predominant serology tests used to diagnose Sjogren’s Disease in community practice settings. We found similar findings in our study cohort. However, although patients with positive ANA and RF and subjective symptoms were given a Sjögren’s diagnosis in the EHR, they were classified as uncertain because they did not have positive SSA antibodies or other confirmatory objective sign tests . Only 10% of the study cohort had a minor salivary gland biopsy and 1% had an ocular staining score and salivary flow test to confirm a diagnosis. Physicians’ easy access to order serology test results and absence of collaborative care with dental providers and ophthalmologists could be the reason for this low use of these tests (lines 416-420).

7. Comment: Line 216

Heroic efforts were performed to train the reviewers.

Response: 

Thank you!

8. Comment: Line 251 and fig 2

441 patients were picked out and 64 patients were excluded resulting in a cohort of 377. 23.9 % positive, 19.6% uncertain and 56.6% negative.

The occurrence of Ig M RF and ANA are the main two laboratory findings that qualify for the patients to be placed into the group of uncertain. Noise?

Response:

There is no noise by choosing rheumatoid factor or ANA to diagnose SD. As you indicated, Sjögren’s may be diagnosed based on various systemic implications. So, this was the reason for placing them in the uncertain category. We have responded to this concern in response to the previous comment 6 for lines 175-177 and table 1.

9. Comment: Line 329

The difference is comorbidity – Is it real or is a consequence of better health care among patients with inflammatory joint or connective disease?

Response: 

We do not know whether the difference in these comorbidities is due to better healthcare delivery. We do know that the severity of the disease varies along with the presence of other comorbid conditions. Also, the difference in the co-morbidities existed only for three medical conditions pulmonary hypertension, hypercholesterolemia, osteoarthritis, across the three SD classifications. 

10. Comment:

Table 5

The most interesting table. Here we see the results of comparison of the two databases.

Comments of the conformity or lack thereof?

And can you speculate if one of the two databases comes closer to the truth? Or are you comparing to databases with great uncertainties?

Response: In the EHR, 377 patients had a diagnosis for Sjögren’s Disease (SD). However, during dental care not all patients in this group reported their Sjögren’s status. The main intent of table 5 is to highlight the gap in the documentation of SD in electronic dental record and the drawback of dentists relying on patients to report their medical conditions. It is not possible for dental providers to access their patient’s medical history without a medical consult. Based on Table 5 results, we conclude the EHR data provides a definite picture of a patient’s medical history especially related to uncommon medical conditions such as SD.

11. Comment: Line 360

It is listed that these criteria could be applicable for classifying SS patients using EHR data for research. It is an assertion that cannot stand alone. For me it is not obviously according to the table 5.

Response: The reviewer seems to be expressing concern based on table 5 results regarding the applicability of criteria (table 1) to classify SD patients using EHR data for research. If that is an accurate assessment of reviewer’s concern, we want to highlight that it is Table 4 and not Table 5 that shows the combinations of symptoms and signs for the classified SD patients using the criteria listed in table 1. Therefore, the conclusion based on these results is that these criteria can be used to identify and classify SD patients using EHR data for research purposes. The significance of table 5 is described in response to comment 10. 

12. Comment: S 373-374

This is the hot spot – Serology test can not be used to diagnose Sjogren or to screen for Sjogren.

Response: 

The existing literature does not report that serology tests cannot be used to diagnose Sjogren’s Disease. Moreover, the American College of Rheumatology (ACR)and European League Against Rheumatism (EULAR) recommend serological tests such as anti-SSA (Ro) or with anti-SSB (La), Rheumatoid factor and ANA as criteria to diagnose SD (Shiboski C et al., 2016). In addition, serology tests are widely used by physicians to diagnose SD (Maciel et al., 2017). Confirmatory objective signs such as salivary gland biopsy, ocular staining test, salivary flow test is rarely performed in a real-world clinical setting. Even the American European Consensus Group classification criteria (Vitali et al., 2002) used for clinical trials recommend serological tests such as anti-SSA or a combined anti-SSA/B to diagnose SD along with the presence of its characteristic cardinal symptoms. 

13. Comment: Line 388-398 – and table 5 and line 403-405 

Dry mouth and dry eyes are symptoms that can be treated symptomatically. But there are many reasons for these symptoms at not necessary a result of an ongoing autoimmune process. I am not familiar with your health care system – could there be an economic incitement to give the diagnoses to at patient. Repay for the bill from the dentist or access to be enrolled into clinical randomized trials for new treatments?

Response: There is no economical incitement for anyone involved. There is only palliative treatment for these patients. There is no cure or a definite treatment for SD patients. The data is obtained from multiple hospitals and researchers do not have any affiliation with them. This is a retrospective study using patient records. It is unfortunate that women get diagnosed very late after the age of 50 and it is a common occurrence that their oral health is affected. We are working on other papers on how oral health is compromised among SD patients. We wanted to bring to light that dentists depend on patient-reported medical conditions to provide oral care. As you would have seen, 69% of the patient records who were diagnosed with SD in the EHR did not have a mention of SD in their dental clinical notes.

14. Comment: Line 403-406 and table 4

The authors are pointing to a very weak point in the daily clinic which are reflected in the records – misdiagnosis or misclassification. Could on of the point in this publication be that there is a urgent need for education.

Response: The possibility of misclassification or misdiagnosis of SD in a clinical setting cannot be ruled out because of the heterogenous and non-specific clinical manifestations during the early stages of SD. However, this study cannot prove this possibility. We have included a sentence on educating clinicians in lines 396-398. Conclusion also has a sentence about educating patients especially women: “there is also the need for screening and awareness of SD disease and its symptoms should be promulgated through educational programs targeting women” (lines 499-502).

Based on our guidelines, patients with no objective findings, and/or with absence of symptoms are deemed negative. Also, understanding what criteria physicians use in their clinical setting to diagnose a patient with Sjögren’s in the absence of objective signs or symptoms warrants further investigation. Finally, as highlighted in the discussion, it is necessary to develop diagnostic criteria other than the classification criteria used for research to diagnose SD in real world clinical settings.

15. Comment: Line 491-492

It is not clear for me how the criteria developed through this study can be applicable in future research that involves early diagnosis. What are the advantages of these versus the EACG criteria?

Response:

Thanks for this comment. We have removed this sentence that this study focused on characterizing clinical findings (lines 499-501). There is a need to develop established criteria to diagnose SD in community clinical practices which is not the intent of this study. AECG criteria focus on recruiting study participants with established signs and symptoms for research studies and clinical trials. It is mandatory to have four of the six criteria (Histopathology or autoantibodies need to be present along with oral, ocular symptoms, oral, ocular signs) and three of the four objective criteria which is not used by physicians to diagnose SD in clinical settings. Our study criteria included any one of the subjective symptoms along with one objective finding, which is typically used to diagnose SD in clinical settings

Reviewer #2: This robust study describes the development criteria for characterizing and classifying findings in electronic health and dental records for Sjogren's Disease patients. This was well written and there are just a few minor comments to consider below.

1. Comment:

Abstract

- As the first study of its kind, the objective misses some opportunity to describe the fullness of the actual study - later in the paper there are 4 aims outlined but it is only written as "characterized" in the abstract.

Response: Thank you. We have revised the abstract to include characterization and association of medical conditions. The revision is done in the abstract (lines 40 to 45)

Original sentence: This study characterized Indiana University School of Dentistry (IUSD) patients’ SD findings using electronic dental record (EDR) and electronic health record (EHR) data available through the state-wide Indiana health information exchange (IHIE).

Revised sentence: This study established criteria to characterize Indiana University School of Dentistry (IUSD) patients’ SD based on symptoms and signs in the electronic health record (EHR) data available through the state-wide Indiana health information exchange (IHIE). Association between SD diagnosis, and comorbidities including other autoimmune conditions, and documentation of SD diagnosis in electronic dental record (EDR) were also determined.

2. Methods

- Consider including a brief explanation as to why those conditions were excluded in the study (line 159-164).

Response: We have now made the following changes: “The above-mentioned conditions are excluded because they mimic clinical symptoms and other characteristic features of SD” (line 168 to 170).

3. Conclusion

- Recommendations are stated but I am curious to how this work could be implemented in the EHR/EDRs at IUSD and beyond?

Response: Implementing interoperability and promoting health information exchange between dental and medical systems at the institutional level is the first step. Educating physicians on the need for diagnostic criteria and patients about the disease condition and the need for screening and multidisciplinary care should be reinforced.

Reviewer #3: 

1. Comment: This is a retrospective study using electronic dental record (EDR) and electronic health record (EHR) data to improve the delayed diagnosis of the patients with Sjögren’s disease (SD) for practicing clinicians. It is interesting because it is unique and highly original, and the research methods and interpretation of the results were conducted without any problems.

However, in order to disseminate our diagnostic criteria widely and clinically, it is necessary to describe in more detail the results compared with the already established standard criteria of SD. I am afraid that the content of this text is insufficient. Please reconsider this point and resubmit it.

Response: Thank you for your positive comments. We want to clarify that the study criteria developed could be used currently only for research studies. None of the classification criteria or the study criteria developed in this study can be used as a diagnostic criterion in clinical settings. The advantage of our developed criteria is it helps characterize the different clinical presentations seen in clinical settings using EHR data and therefore, can be used to generate study cohorts for longitudinal studies that has not been previously studied. We have highlighted this difference in the introduction (lines 120 to123) and is the rationale for this study. . Also, we cannot compare the AECG criteria with our study criteria because the AECG criteria is used to establish diagnosis before enrolling participants in prospective studies. In contrast, this study criteria can only be used to characterize SD patients’ clinical findings using EHR data for research purposes. Further studies are required to determine the appropriateness of using these criteria to diagnose SD during patient care.

Reviewer #4:

1. Comment: Thank you for this manuscript. You used a lot of data to find more clearity in the SD diagnosis mainly in dental health records. You do present a lot of numbers and results however the conclusions based on these numbers are somewhat vague to me. What do you conclude? And what are the results concerning your aims? For me it reads like a relatively large part of the patients most likely do not have SD (as they also have rheumatiod arthritis or lupus). And what does the EDR data add to the EHR data. It is not clear for me from the results and discussion.

To me it feels like the results are not as positive as the authors hoped for. And therefore, the results are kept a bit vague. Even if the result is not what you hoped for you should report this clearly. A 'negative' result is also a result.

Response: This is not a negative study. It is a descriptive study. The goal, which we achieved, was to develop criteria to identify patients with an inconsistently diagnosed condition using their EHR data and NOT their dental record data. We cast a wide net in identifying patients to initially include, and the criteria we developed narrowed the set of patients to those who are highly like to have SD. Co-occurring medical and autoimmune conditions are common among these patients that causes difficulty in diagnosing them. Also, patients with a SD diagnosis having a second autoimmune condition such as rheumatoid arthritis or lupus are considered to have secondary SD. In this study, 53% of the study cohort had a second autoimmune condition and can be considered to have secondary SD. Tables 5 and 6 show how many SD patients reported their SD condition during dental care and how many of them had it diagnosed before and after their dental care. We added this information because dentists always rely on patient-reported medical conditions to provide care. We state this in the introduction (4th paragraph line 75 & 116) and mention in the discussion the gaps in the documentation of SD diagnosis during dental visits (lines 448 to 471). This information and data are added to highlight the importance of sharing medical and dental information.

 References:

Shiboski SC, Shiboski CH, Criswell LA, Baer AN, Challacombe S, Lanfranchi H, et al. American College of Rheumatology classification criteria for Sjögren’s syndrome: A data-driven, expert consensus approach in the Sjögren’s International Collaborative Clinical Alliance Cohort. Arthritis Care & Research. 2012;64(4):475–87. 

Maciel G, Crowson CS, Matteson EL, Cornec D. Prevalence of Primary Sjögren’s Syndrome in a US Population-Based Cohort. Arthritis Care Res (Hoboken). 2017/08/31 ed. 2017;69(10):1612–6.

Vitali C, Bombardieri S, Jonsson R, Moutsopoulos HM, Alexander EL, Carsons SE, et al. Classification criteria for Sjögren’s syndrome: a revised version of the European criteria proposed by the American-European Consensus Group. Ann Rheum Dis. 2002/05/15 ed. 2002 Jun;61(6):554–8.

---

## [Decision Letter · Decision Letter 1]

18 Jul 2023

Characterizing Clinical Findings of Sjögren’s Disease Patients in Community Practices Using Matched Electronic Dental-Health Record Data.

PONE-D-22-35694R1

Dear Dr. Thyvalikakath,

We’re pleased to inform you that your manuscript has been judged scientifically suitable for publication and will be formally accepted for publication once it meets all outstanding technical requirements.

Kind regards,

Boyen Huang, DDS, MHA, PhD

Academic Editor

PLOS ONE

Additional Editor Comments (optional):

Reviewers' comments:

Reviewer's Responses to Questions

**Comments to the Author**

1. If the authors have adequately addressed your comments raised in a previous round of review and you feel that this manuscript is now acceptable for publication, you may indicate that here to bypass the “Comments to the Author” section, enter your conflict of interest statement in the “Confidential to Editor” section, and submit your "Accept" recommendation.

Reviewer #2: All comments have been addressed

Reviewer #3: All comments have been addressed

Reviewer #4: All comments have been addressed

2. Is the manuscript technically sound, and do the data support the conclusions?

Reviewer #2: Yes

Reviewer #3: Yes

Reviewer #4: Yes

3. Has the statistical analysis been performed appropriately and rigorously? 

Reviewer #2: Yes

Reviewer #3: Yes

Reviewer #4: Yes

4. Have the authors made all data underlying the findings in their manuscript fully available?

Reviewer #2: Yes

Reviewer #3: Yes

Reviewer #4: Yes

5. Is the manuscript presented in an intelligible fashion and written in standard English?

Reviewer #2: Yes

Reviewer #3: Yes

Reviewer #4: Yes

6. Review Comments to the Author

Reviewer #2: All of my comments and those of other reviewers were addressed. Excellent manuscript and study on to address the community practices of clinical findings of Sjögren’s Disease Patients in matched electronic records.

Reviewer #3: I have confrmed the revise by authors according to my comment.

so, there are no further corrections to be made.

Reviewer #4: (No Response)

7. PLOS authors have the option to publish the peer review history of their article (what does this mean?). If published, this will include your full peer review and any attached files.

Reviewer #2: No

Reviewer #3: No

Reviewer #4: No

---

## [Editor Report · Acceptance letter]

21 Jul 2023

PONE-D-22-35694R1 

Characterizing Clinical Findings of Sjögren’s Disease Patients in Community Practices Using Matched Electronic Dental-Health Record Data. 

Dear Dr. Thyvalikakath:

I'm pleased to inform you that your manuscript has been deemed suitable for publication in PLOS ONE. Congratulations! Your manuscript is now with our production department. 

Kind regards, 

on behalf of

Dr Boyen Huang 

Academic Editor

PLOS ONE